# Concurrent Bering Sea and Labrador Sea ice melt extremes in March 2023: A confluence of meteorological events aligned with stratosphere-troposphere interactions

Thomas J. Ballinger[1], Kent Moore[2,3], Qinghua Ding[4], Amy H. Butler[5], James E. Overland[6], Richard L. Thoman[1], Ian Baxter[4], Zhe Li[4], and Edward Hanna[7]

[1]International Arctic Research Center, University of Alaska Fairbanks, Fairbanks, AK, USA
[2]Department of Physics, University of Toronto, Toronto, Ontario, Canada
[3]Department of Chemical and Physical Sciences, University of Toronto, Canada, Toronto, Ontario, Canada
[4]Department of Geography, and Earth Research Institute, University of California, Santa Barbara, Santa Barbara, CA, USA
[5]Chemical Sciences Laboratory, National Oceanic and Atmospheric Administration, Boulder, CO, USA
[6]Pacific Marine Environmental Laboratory, National Oceanic and Atmospheric Administration, Seattle, WA, USA
[7]Department of Geography and Lincoln Climate Research Group, University of Lincoln, Lincoln, UK

*Correspondence to*: Thomas J. Ballinger (tjballinger@alaska.edu)

**Abstract.** Today's Arctic is characterized by a lengthening of the sea ice melt season, but also by fast and at times unseasonal melt events. Such anomalous melt cases have been identified in Pacific and Atlantic Arctic sector sea ice studies. Through observational analyses, we document an unprecedented, concurrent marginal ice zone melt event in the Bering Sea and Labrador Sea in March of 2023. Taken independently, variability in the cold season ice edge at synoptic time scales is common. However, such anomalous, short-term ice loss over either region *during the climatological sea ice maxima* is uncommon, and the tandem ice loss that occurred qualifies this as a rare event. The atmospheric setting that supported the unseasonal melt events was preceded by a sudden stratospheric warming event amidst background La Niña conditions that led to positive tropospheric height anomalies across much of the Arctic and the development of anomalous mid-troposphere ridges over the ice loss regions. These large-scale anticyclonic centers funneled extremely warm and moist airstreams onto the ice causing melt. Further analysis identified the presence of atmospheric rivers within these warm airstreams whose characteristics likely contributed to this bi-regional ice melt event. Whether such a confluence of anomalous wintertime events associated with troposphere-stratosphere coupling may occur more often in a warming Arctic remains a research area ripe for further exploration.

## 1 Introduction

Observational analyses of the Arctic atmosphere have noted warmer air temperatures and increased moisture content during the last two decades relative to previous years (Ballinger et al., 2023; Boisvert et al., 2023). Periods of increased climate variability (Hanna et al., 2015) can coincide with these atmospheric changes in the Arctic to produce extreme meteorological

phenomena, which may influence human and environmental systems both within and beyond the high northern latitudes. Moreover, terrestrial Arctic snow and sea ice extent, area, and depth/thickness control heat exchange between the land, ocean, and atmosphere (Serreze and Barry, 2011). With less snow and sea ice in a warming Arctic, instances of surface-to-atmosphere heating perturbations can magnify impacts of synoptic circulation patterns on local and/or remote surface weather extremes (Francis and Vavrus, 2015; Zhang et al., 2018; Tachibana et al., 2019; Bailey et al., 2021). Thus air-sea interactions resulting in extreme events in today's Arctic are structurally complex (Walsh et al., 2020) and shaped by the surface condition/type and prevailing weather pattern (Overland et al., 2021).

A key consideration of complex Arctic extreme events is their timing of occurrence within the annual cycle. As an example, the Arctic Ocean's ice cover tends to thin and decline (thicken and increase) through the boreal summer (winter) months up to the September minima (March maxima). However, analyses of satellite observations have shown a trend toward earlier melt onset across most of the Arctic marginal seas (e.g., Stroeve and Notz, 2018) with unusually-timed and often isolated ice loss events during winter or early spring interspersed on these trends. The North Atlantic Arctic region that includes marginal seas around Greenland, Iceland, and northwest Europe has experienced several of these cases in recent times. During mid-April of 2013, a persistent anticyclone over Greenland coincided with record-early melt onset in the Baffin Bay, Davis Strait, and Labrador Sea region that was ~8 weeks earlier than the 1981-2010 average (Ballinger et al., 2018). Above freezing air temperatures at the North Pole during late December of 2015 led to a substantial loss of sea ice over the Arctic Ocean (Moore, 2016). In late February and early March of 2018, a polynya unexpectedly opened off the northern Greenland coast that was driven by anomalously warm and strong southerly winds that were preceded by a sudden stratospheric warming (SSW) event (Moore et al., 2018). In one of the most notable examples, an Arctic cyclone that registered record-low central pressure traversed the Barents and Kara seas in late January of 2022 and caused record surface winds and attendant ice loss for the time of year (Blanchard-Wrigglesworth et al., 2022). Unlike the previous cases, dynamical and ocean processes rather than thermodynamics were attributed to this unseasonal ice loss event.

There is a large body of research into so-called compound extreme climate events such as the simultaneous occurrence in a particular region of a drought and heat wave or a storm surge and fluvial flooding (e.g., Zscheischler et al., 2018; AghaKouchak et al., 2020). Less well-studied are so-called concurrent climate extreme events where two or more spatially isolated regions are subject to simultaneous or near-simultaneous extremes (Zhou et al., 2023). Compound events may be associated with a single overarching phenomenon such as a hurricane, while concurrent events are typically associated with amplified Rossby Waves (Kornhuber et al., 2020).

In this study, we have identified the first known observation of a concurrent climate extreme event in the Arctic as well as one that is associated with a SSW and La Niña background state. This concurrent event is marked by unusually-timed sea ice melt in the Bering Sea and Labrador Sea during March of 2023. Our goals in this observationally-based case study are to describe

the respective regional sea ice conditions during March 2023, place them in historical spatial and temporal context, and
evaluate the synoptic atmospheric mechanisms responsible for the ensuing melt extremes. As part of our analyses, we evaluate
the probability of such sea ice melt extremes amidst the period that encompassed the climatological Arctic sea ice maximum.
We conclude with a discussion of our findings that considers seasonal and synoptic meteorological anomalies that occurred
during and around the time of these melt events. Our conclusions also touch upon the implications of Arctic warming for
analogous future melt events.

## 2. Data and Methods

*2.1 Sea ice and atmospheric datasets*

Daily sea ice concentration (SIC in %) is derived from the NOAA/NSIDC Climate Data Record (CDR) of passive microwave
SIC, version 4 (Meier et al., 2021, 2022). This dataset represents a blended product of the NASA Team algorithm (Cavalieri
et al., 1984) and NASA Bootstrap algorithm (Comiso, 1986), and is available daily on a 25 km$^2$ grid from 1979-onwards.
ECMWF fifth generation global atmospheric reanalysis (ERA5) data at their 31 km native resolution for 1979-2023 (Hersbach
et al., 2020) are used to evaluate atmospheric conditions across the Arctic region during and around the SSW event and ensuing
sea-ice melt extremes. ERA5 fields examined include 2-meter air temperature (T2m in °C), total column water vapor (in mm),
total precipitation, which is the sum of large-scale and convective precipitation including rain and snowfall, that reaches the
surface (in mm/day), net and downward longwave radiation (in W/m$^2$), and geopotential heights (in m) over the atmospheric
column from 1000 hPa to 1 hPa. Unless otherwise stated, data are binned to daily means. Studies have shown ERA5 to be
effective at capturing Arctic weather and climate variability. As an example, during a research expedition in Fram Strait,
Graham et al. (2019) noted ERA5 air temperatures, humidity, and winds exhibited relatively strong correlations and low biases
in comparison with radiosonde observations and performed better overall than other modern atmospheric reanalyses in the
region. Numerous other studies have relied upon ERA5 data to understand the synoptic evolution and characteristics of
airstreams within the Arctic (e.g., Nygard et al., 2020; Papritz et al., 2022; Kirbus et al., 2023).
In addition to reanalysis fields, daily averaged T2m data from regional weather stations are evaluated (**Figure 1**). We
deliberately selected near-coastal weather stations based on several criteria, including multidecadal records that are relatively
complete (>95% of dates surveyed register a T2m value) for sites located north and south of both the early March long-term
mean and 2023 ice edge in the Bering Sea and Labrador Sea, respectively. Data from leap years are omitted as 2023 was not
one. For the Bering Sea region, we obtained T2m data from the National Centers for Environmental Information Applied
Climate Information System (NCEI ACIS) for Alaska terrestrial weather stations at St. Paul (57.16°N, 170.22°W) and
Kotzebue (66.89°N, 162.58°W). The St. Paul historical record is surveyed from 1916-2023, while the Kotzebue record is
assessed from 1923-2023. For the Labrador region two western Greenland weather station records, which are maintained by
the Danish Meteorological Institute (DMI), are obtained for Nuuk (64.17°N, 51.75°W) and Aasiaat (68.70°N, 52.75°W). Both
of these Greenland records span 1958 to 2023. We supplement NCEI ACIS and DMI observations with Programme for
Monitoring of the Greenland Ice Sheet (PROMICE) automatic, on-ice weather station temperatures, measured from a nominal
height of 2.7 m above the ice-sheet surface, for two sites: one is near Nuuk on a peripheral glacier (NUK_K; 64.16°N ,
51.36°W; 710 m asl) and the other is found within the lower ablation area of the Greenland Ice Sheet (GrIS) in the Qassimiut
region (QAS_L; 61.03°N, 46.85°W; 280 m asl) (Fausto et al., 2021). The PROMICE data records are relatively short, with
NUK_K established in 2015 and QAS_L in 2008, though both are 99% complete for the dates we surveyed and provide
valuable information on GrIS in situ air temperatures on the rather observationally sparse Greenland Ice Sheet.

Several atmospheric indices are analyzed and discussed in this work. The SSW compendium (Butler et al., 2017, updated), a
long-term archive of indicator climate indices associated with SSW events, confirmed the onset of the late-winter 2023 SSW
event (16 February). We examine one such metric of this archive that we term the Polar Vortex Index (PVI) that describes the
daily-mean, zonal-mean winds at 60°N and 10 hPa, where the timing of the shift from westerly to easterly stratospheric flow
between November and April signifies the SSW onset (Charlton and Polvani, 2007). The PVI is analyzed from 1979-2023.
SSWs are known to influence the mid-to-high latitude tropospheric circulation patterns and often precede a negative North
Atlantic Oscillation (NAO) regime and high-latitude anticyclonic blocking (Baldwin et al., 2021). Therefore, we elect to
analyze the daily NAO and region-specific Greenland Blocking Index (GBI) and Alaska Blocking Index (ABI). The NAO
used here extends from 1950 to 2023 and is defined as the leading, rotated principal component of standardized 500 hPa
geopotential height (z500) anomalies from 20-90°N (Barnston and Livezey, 1987). The GBI describes the mean z500 across
60-80°N, 20-80°W (Hanna et al., 2013), and the ABI depicts the averaged z500 from 55-75°N and 125-180°W (Ballinger et
al., 2022). These blocking indices are analyzed over the 1948 to 2023 period.

*2.2 Extreme event detection methods*
We examine moisture transport into the Arctic during our case study by employing an atmospheric river (AR) detection
algorithm developed by Guan and Waliser (2019). This algorithm is applied on 6-hourly ERA5 integrated water vapor transport
(IVT in kg/m/s) data, averaged from 1000hPa to 300 hPa on a 1.5° x 1.5° global grid. In this framework, ARs for each 6-hour
interval are defined when an IVT threshold exceeding the 85th percentile of climatological IVT is reached for a grid cell in the
domain of interest. Additionally, these ARs must meet specific criteria related to the orientation, length, and length-to-width
ratio of IVT, as outlined by Guan and Waliser (2019). Widely adopted in previous studies spanning the tropics to the high
latitudes including the Arctic and Antarctic, this algorithm serves as a reliable scheme for AR analysis (Collow et al., 2022).
We examine the duration of AR events passing through the Alaska and Greenland regional domains shown in **Figure 1** leading
up to, coinciding with, and following the Bering and Labrador melt events, respectively. AR duration is defined as the
percentage (%) of the day in which an AR resides within any portion of the respective domains. We also measure the intensity
of AR events, defined here as the mean IVT of all grid cells that cross into either domain associated with an AR.

Daily atmospheric indices and maps of the reanalysis data are presented, and values are identified that meet or exceed an

extreme value threshold (i.e., 95[th] or 99[th] percentile) relative to a specified number of days across the data records described in

Section 2.1. For example, a 99[th] percentile St. Paul, Alaska T2m value during the 90-day "winter" period from 1 January – 31

March 1916-1923 (where 9576 days registered a daily mean T2m reading) is 3.3°C. Use of the full historical period or select

portions of dataset's records along with extended time windows (e.g., 1 January – 31 March) provided a larger sample size

from which to calculate extreme values relative to the period specified or season (e.g., 90 days) versus a singular date of

reference.

## 3. Results

*3.1 Extreme and unusually-timed sea ice melt*

The regional SIC means, variability, and anomalies around the peak of the melt events relative to 1-15 March 2000 to 2023

are shown in **Figure 1**. This subset of years is selected as winter months since 2000 have seen a large decline in sea ice

conditions (Stroeve and Notz, 2018). In the Labrador region, the 50% climatological ice edge tilted northeast to southwest

from Davis Strait into the Labrador Sea and transitioned in the marginal ice zone to nearly 100% SIC on the western flank of

this boundary (**Figure 1a**). In contrast, the Bering Sea ice edge exhibited a more zonal orientation and extended from ~61°N

in the western Bering Sea to ~59°N in the eastern Bering Sea (**Figure 1d**). From 2000 to 2023, interannual SIC variability for

the first half of March in these marginal ice zone areas was ~30% (**Figure 1b, e**), while early March 2023 saw SIC reductions

along the ice edge on the order of ~-30% (**Figure 1c,f**).


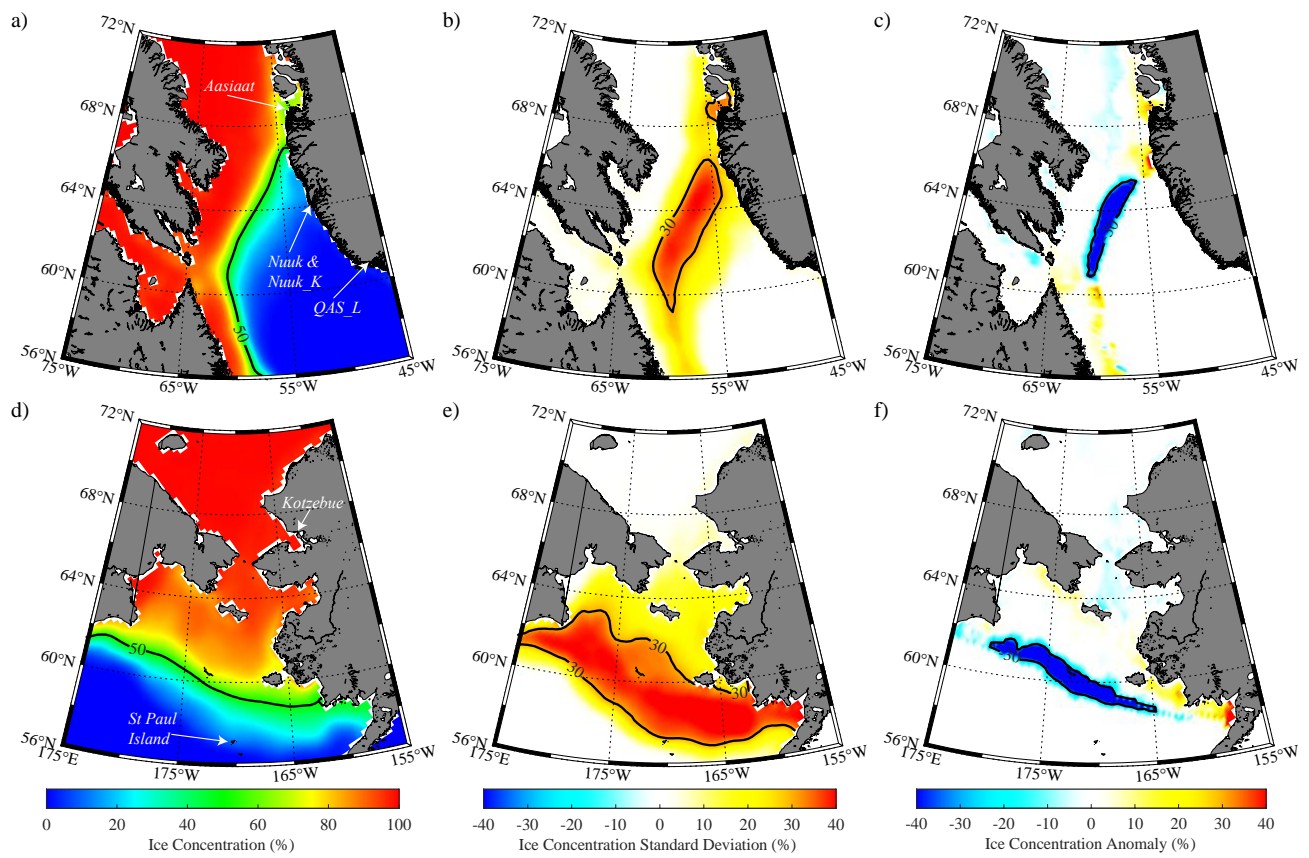

**Figure 1.** Sea ice concentration (SIC in %) from the NOAA/NSIDC CDR dataset. Mean conditions for the period 1-15 March 2000-2023 for: (a) the Labrador Sea and d) the Bering Sea. The SIC standard deviation (%) for 1-15 March 2000-2023 is shown for: (b) the Labrador Sea and e) the Bering Sea. The sea ice concentration anomaly on 5 March 2023 relative to the 1-15 March 2000-2023 period is shown for (c) the Labrador Sea and (f) the Bering Sea. In (a) and (d) locations of the weather stations mentioned in the text are indicated with arrows.


The SIC conditions in these areas of >30% variability are examined more closely with respect to the winter of 2023. Winter is
loosely defined here as January through March. From mid-January through February, the daily Labrador SIC exceeded the
2000-2023 mean, then abruptly plummeted to below-normal conditions in early March and remained below-average through
the end of the month (**Figure 2a**). The Bering SIC showed more variability about the SIC day-of-year means with periods of
slightly above and below-normal ice cover into early March and through the rest of the month (**Figure 2b**). While single day
SIC departures through winter in both areas did not breach the 5[th] or 95[th] percentiles for the day of year, the largest 4-day
changes (<20% SIC losses) occurred roughly at the same time and culminated on March 5[th] in the Labrador Sea and March 6[th]
in the Bering Sea (see dashed red vertical lines in **Figure 2**). While day-to-day sea ice variability is not unusual throughout
winter, the day-of-year mean curves (thick black lines in **Figure 2**) suggest that ice growth tends to continue in both of these
regions throughout most of March aligned with the typical pan-Arctic sea ice maximum (Meier et al., 2023).

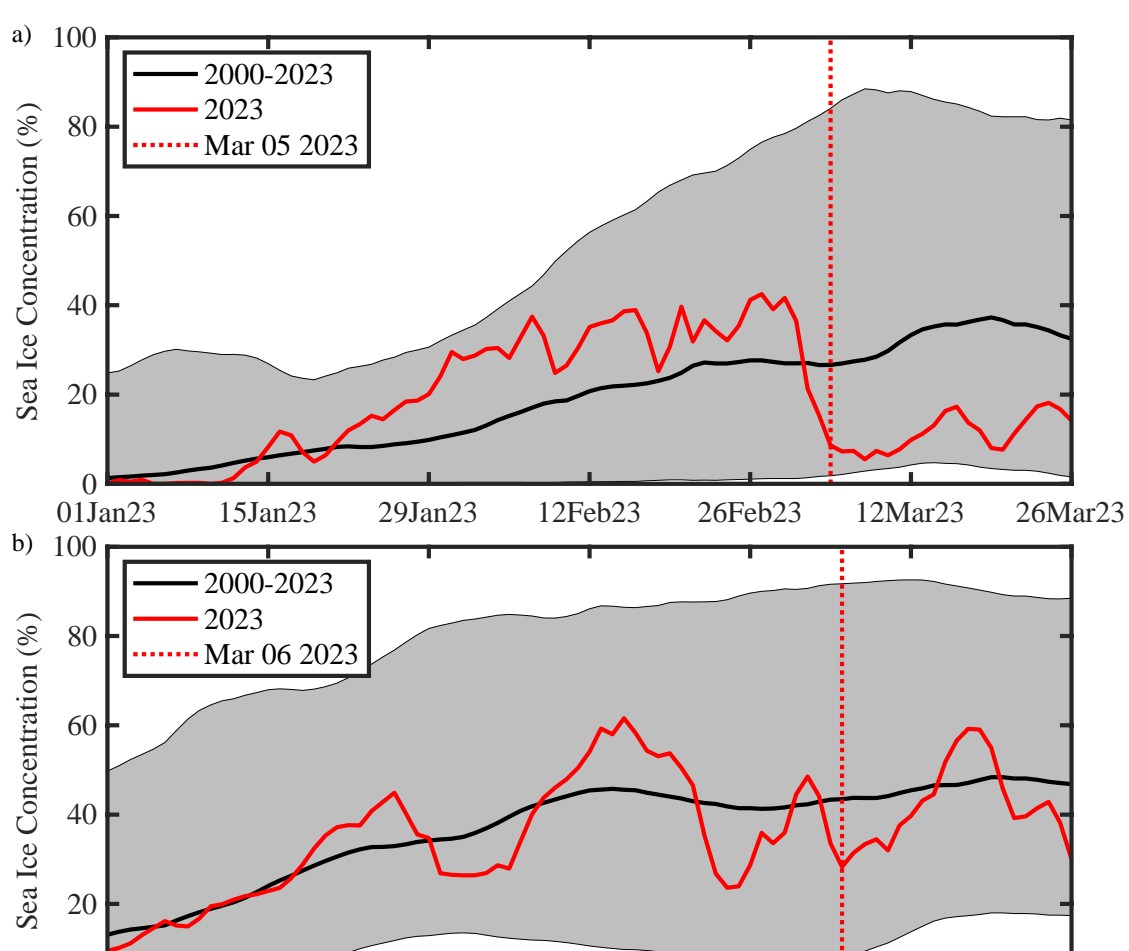

**Figure 2.** Time series (red curves) of the daily SIC averaged over the regions, a) Labrador Sea and b) Bering Sea, respectively in Figures 1b,e, where the standard deviation exceeds 30% for the period 1 January to 26 March 2023. The black line represents the daily mean value for the period 2000-2023 with the shading incorporating daily values between the 5th and 95th percentile values. The ending dates for the 4-day window with the largest change in ice concentration are shown with the dotted red lines.


Histograms provide additional probabilistic perspective on the likelihood of such 4-day ice loss events for the times of year
they occurred in 2023 (**Figure 3**). Since 2000, both the Labrador Sea (red curve) and Bering Sea (blue curve) have shown
quasi-normal SIC distributions over the 1-15 March period. The 2023 4-day changes in both areas, characterized by ~20% SIC
reduction in the Bering Sea and ~27% SIC decline in the Labrador Sea, represent extreme outliers found in the far-left tails of
their respective data distributions. The magnitude of these short-term SIC loss events is uncommon for the time of year, which
prompts further investigation into the synoptic processes that drive, and potentially link, these rare, concurrent events.

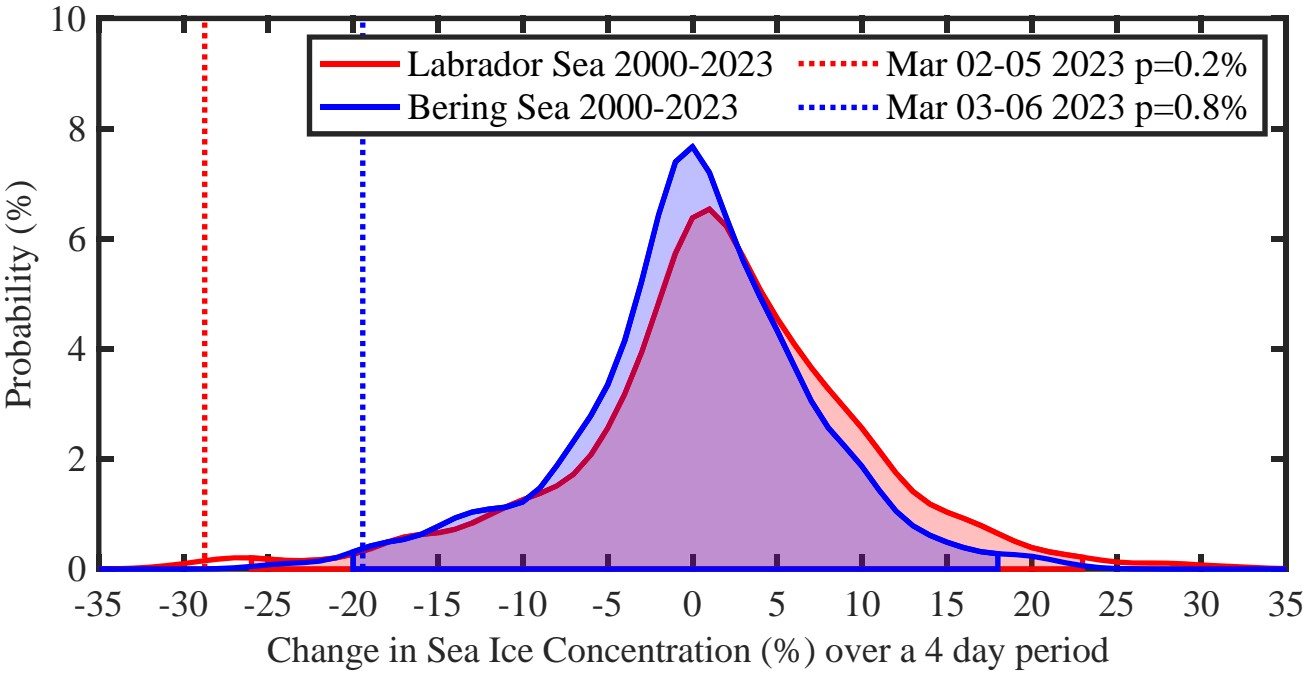

**Figure 3.** Histograms of the change over a 4-day period during 1-15 March 2000-2023 for the Labrador Sea (red) and Bering Sea (blue) regions used in Figure 2. The shading represents the regions bounded by the 1st and 99th percentile values. The largest changes during March 2023 are indicated by the dashed lines.

*3.2 Synoptic mechanisms, part 1: The 2023 SSW event and its stratosphere-troposphere signatures*
On 16 February, a SSW occurred that appears to have strongly contributed to the synoptic environment in early March that led
to the cross-Arctic melt events. **Figure 4** shows the winter-long evolution of the height anomalies with respect to the SSW
event. In mid-January 2023, positive tropospheric heights in the 1000-100 hPa layer preceded positive height anomalies aloft
that developed toward late January and early February. The positive height anomalies indicate upward troposphere to
stratosphere coupling that resulted in a minor stratospheric warming event at the end of January. Over the two weeks that
followed, a second, stronger and positive (~2 sigma) geopotential height anomaly developed aloft within the upper stratosphere
and peaked on 16 February in conjunction with the day of the shift from westerly to easterly 10 hPa winds at 60°N found in
the PVI (**Figure 5a**), which marked the date of SSW onset (Butler et al., 2017, updated). The PVI dipped to roughly the 1[st]
percentile following SSW onset on 28 February and 1 March, characterizing this as an anomalously strong event for this time
of year. The PVI reached a minimum wind speed of -18 m/s on 28 February, which places it as the 6[th] strongest reversal (out
of 28 such events) of the polar vortex winds during a stratospheric warming from 1979-2023 (Lee and Butler 2019). As is the
tendency with SSWs, the influence of the above-average, upper stratospheric air pressures and temperatures (latter not shown)
descended during this time, yielding increased heights across the depth of the stratosphere through late February (**Figure 4**).
By early March, the SSW warming signal propagated toward the surface and large positive height anomalies extended through
the depth of the tropospheric column. The largest positive height anomalies within the lower troposphere and at the surface
manifested predominantly over the Greenland/Labrador Sea area (**Figure S1**), as is typical following SSW events (Baldwin et
al., 2021), and the timing coincided with the Bering Sea and Labrador Sea melt events.

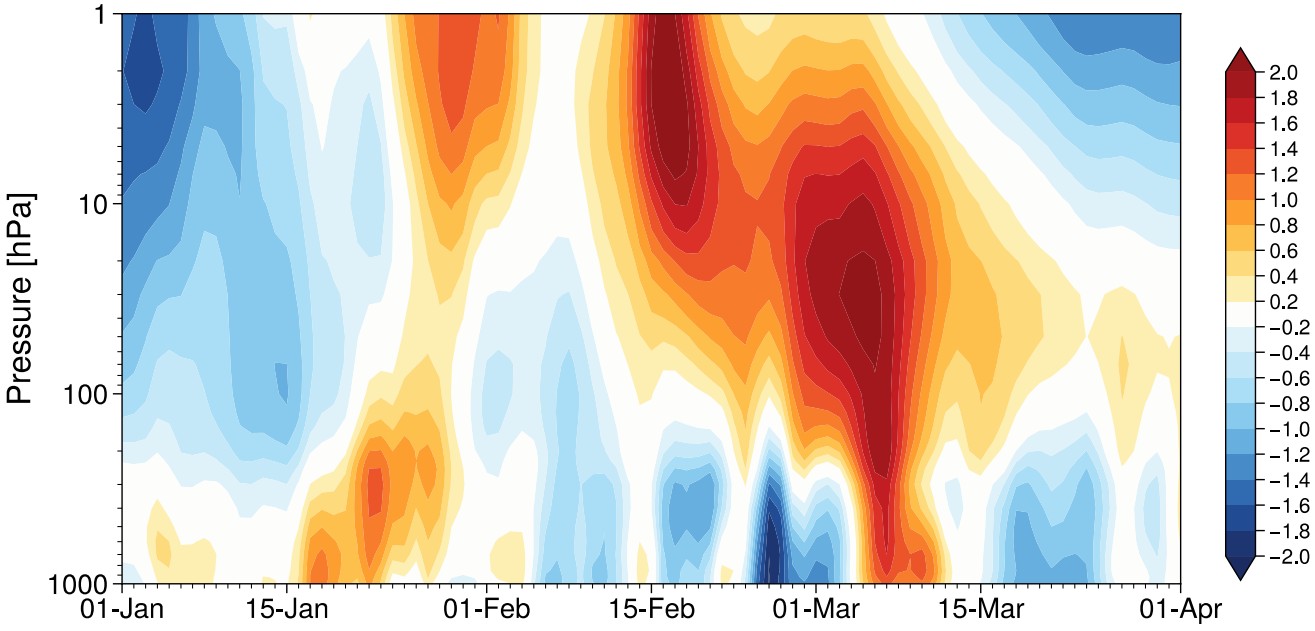



**Figure 4.** Polar cap (60-90°N) standardized geopotential height anomalies (unitless) from the surface to the upper stratosphere
during winter 2023. The standardized anomalies are calculated at each pressure level by removing the daily climatology and
dividing by the daily standard deviation. The standardized anomalies are shown relative to the day of year for the 1979-2023
period of the ERA5 reanalysis.

In the two weeks that led up to this strong SSW event, the large-scale mid-tropospheric circulation was characterized by a positive NAO fluctuation between 0 and 1.5 sigma, indicative of stronger than normal westerly winds across the mid-to-high latitudes (**Figure 5b**). Negative height anomalies (lower than normal pressure) across most of the polar cap troposphere between 1-15 February (**Figure 4**) support this assertion. After the SSW event on 16 February, the NAO slightly increased for two days then plummeted, reversed sign, and became strongly negative (~-1 sigma) from 2-8 March around the melt events (**Figure 5b**). Zooming in on the study regions of interest, strong, lagged ridging responses are noted in the respective mid-tropospheric height fields. The z500 pattern atop the Labrador Sea area of ice loss described by the GBI is >100 m above-average from 1-12 March, including record-high day of year departures (since 1948) from 4-7 March when the GBI values exceeded the 99th percentile (**Figure 5c**). This period also corresponded with the strongest downward coupling of the SSW event to surface conditions (**Figure 4**). While comparatively not as extreme as those of the GBI, ABI values are also considerably higher-than-average during most of the same period (4-12 March), punctuated by >100 m anomalies from 5-11 March (**Figure 5d**). These higher-than-average ABI values appear related to a persistent high pressure system over the broader North Pacific region – a signature of the La Niña extratropical teleconnection – that moved into and out of the Alaskan region on synoptic timescales during the January to March period. As a possible precursor to the SSW event and subsequent pressure increase through much of the atmospheric column, initially there were ABI peak with values >150 m above-normal from 27-30 January capped by the 28 January ABI value (5496.83 m) falling just shy of the 99th percentile (5501.82 m). The Alaska ridge (not shown) associated with these elevated ABI values appears well-timed with upward coupling of the troposphere to the stratosphere (**Figure 4**). We revisit this discussion in **Section 4**.

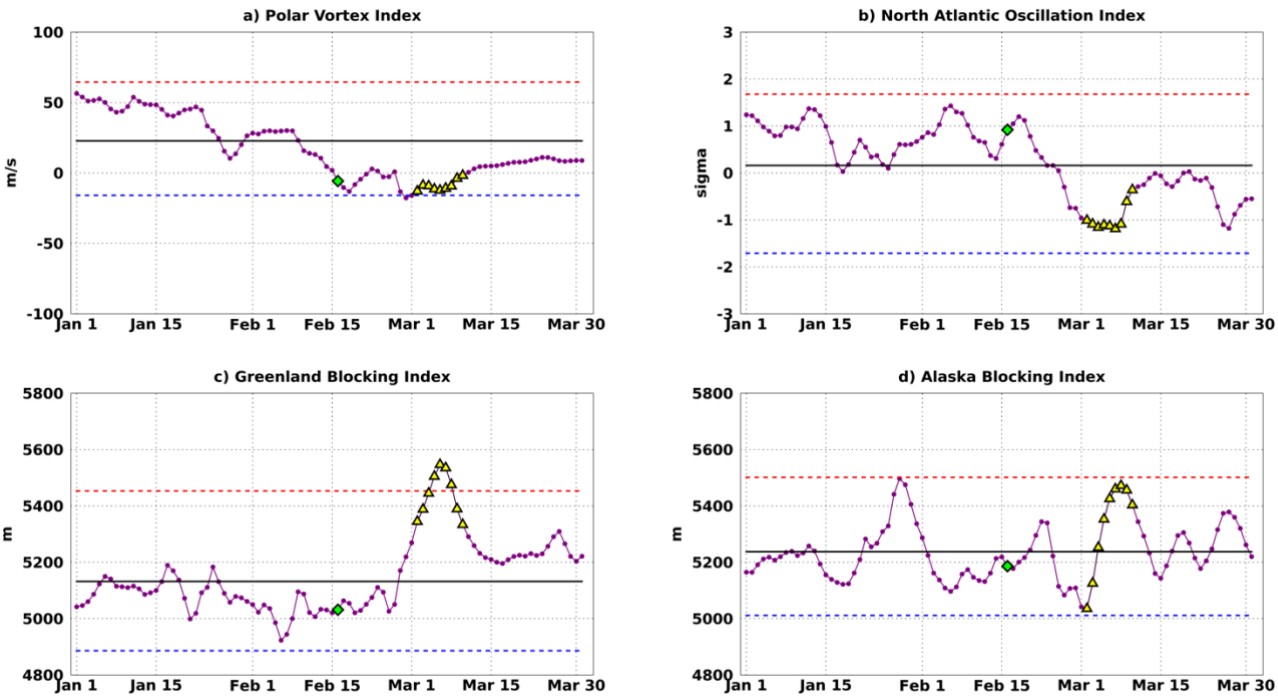

**Figure 5.** Daily atmospheric indices for 1 January – 31 March 2023 (purple lines) overlapping the multi-sectoral melt event for the a) Polar Vortex Index (m/s), b) North Atlantic Oscillation Index (standardized), c) Greenland Blocking Index (m), and d) Alaska Blocking Index (m). Considering all days from 1 January to 31 March for the respective indices' full periods of record (see Section 2.1), the mean of each variable (black line), 1st percentile (blue dashed line), and 99th percentile (red dashed line) are shown in each graphic. The sudden stratospheric warming event on 16 February 2023 is labeled with a green diamond, and to draw attention to the dates around the Labrador Sea and Bering Sea melt events, the period from 2-10 March 2023 is identified by yellow triangles.

215

The evolution of the day-to-day z500 spatial pattern in March provides perspective to the values of the large-scale circulation and regional blocking indices overlapping the melt events. The height pattern over Greenland, Baffin Bay and Labrador Sea is above-normal and successively strengthens during 2-4 March (**Figure 6a-c**) before the peak in the short-term Labrador Sea melt observed on 5 March when western Greenland and Baffin Bay is engulfed in >99[th] percentile height anomalies (**Figure 6d**). Meanwhile, below-normal mid-tropospheric pressure over Alaska and poleward of the central Bering Sea from 2-4 March gave way to higher-than average pressure by 5 March and preceded the 6 March peak in the Bering Sea ice loss (**Figure 6e**). A large-scale dipole structure is evident from 6-10 March, as the North American (Eurasian) high-latitudes spanning the International Dateline (i.e., 180°W) to ~30°W (30°W-180°W) exhibited higher-than-normal (lower-than-normal) heights with extreme departures around Greenland (**Figure 6e-i**) that are reflected in the magnitude of the daily GBI anomalies (**Figure 5c**). Midtropospheric ridging over high-latitude North America with larger anomalies over Greenland than Alaska represents a common regional weather regime (Lee et al., 2023), however, the z500 anomalies observed during the latter portion of our case study are relatively higher in magnitude. In terms of set-up, over the 9-day period, the blocking pattern developed initially over the Iceland region before retrograding westwards over Greenland towards the Labrador Sea and Baffin Bay. Such retrograde movements have been noted to occur in other cases of blocking development over the Greenland region (Hanna et al., 2018). While the z500 pattern orientation and development are not uncommon, the strength of the anticyclonic anomalies is notable in this case.

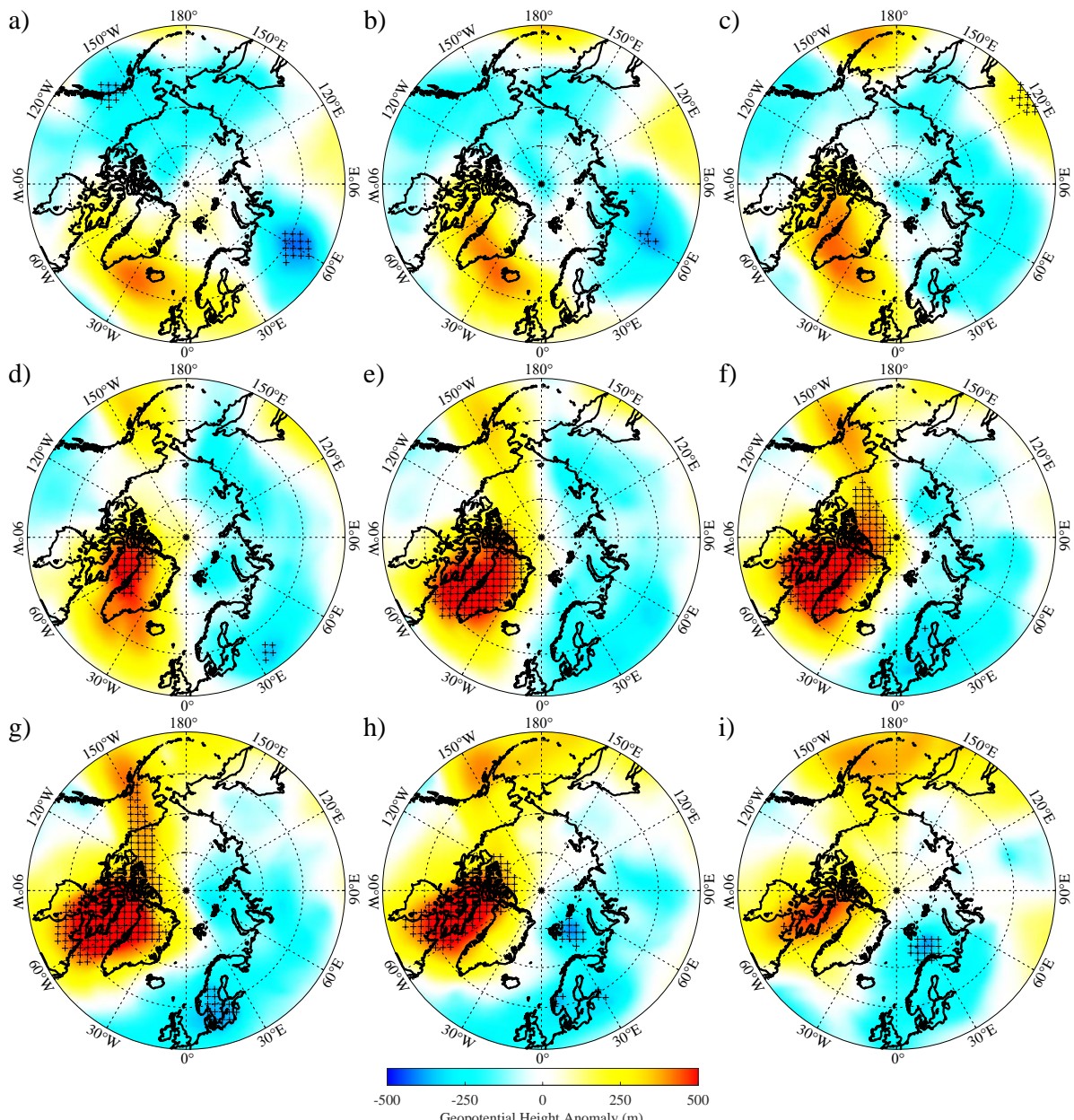

233

234

**Figure 6**. 500 hPa geopotential height (z500) anomaly (m) from the ERA5 at 0 GMT on: a) 2 March, b) 3 March, c) 4 March, d) 5 March, e) 6 March, f) 7 March, g) 8 March, h) 9 March, and i) 10 March 2023. The anomalies are presented with respect to the period 16 February – 15 March 1979-2023. Gridpoints where the anomalies are less than the 1st percentile (blue hues) or greater than the 99th percentile (red hues) based on the above period are indicated with the '+'.


*3.3 Synoptic mechanisms part 2: Thermodynamic effects*
In the following, we examine the thermodynamic environment overlapping the aforementioned atmospheric circulation
anomalies. **Figure 7** shows the daily pan-Arctic T2m anomaly field (shading) around the melt events; the 0°C isotherm (blue
contour) is overlaid for reference. During 2-4 March, air temperature anomalies over south central Greenland, Davis Strait,
and northern Labrador waters overlapping the ice edge were above-normal (**Figure 7a-c**). In particular, from the 2nd to the 3rd
of March, the 0°C isotherm abruptly migrated westward and encompassed much of the Labrador Sea including the ice edge
(refer to **Figure 1a**). During this time 99th percentile warm extremes were found across the northern Labrador Sea, the southern
tip of Greenland, and the southwestern Irminger Sea. Warm extremes persisted in the vicinity of the ice edge on 5 March
(**Figure 7d**), then the large temperature anomalies (~15-16°C) expanded to cover much of the area from the Labrador Sea
through Baffin Bay on 6-7 March (**Figure 7e,f**). While the warm air mass appeared to propagate westward into northeastern
Canada in the days that followed, T2m anomalies remained above-average in these areas until colder air moved into the region
on 10 March (**Figure 7g-i**).

A warm air incursion into the Bering Sea was also apparent during this same time. From the 3rd to the 4th of March, the 0°C
isotherm migrated several degrees northward as anomalously warm air penetrated into the Bering region (**Figure 7b,c**). The
general southwest to northeast trajectory of the mild airstream was apparent in the days that followed. The 0°C isotherm entered
the northeastern Bering Sea and southwestern Alaska on 5 March as anomalous melt along the ice edge continued, while
temperatures over the western Bering Sea and northeastern Siberia remained below normal (**Figure 7d,e**). Air temperatures
remained above average to extreme in western and northern Alaska during the days that followed as the airmass propagated
into the high Arctic over 6-10 March (**Figure 7f-i**).

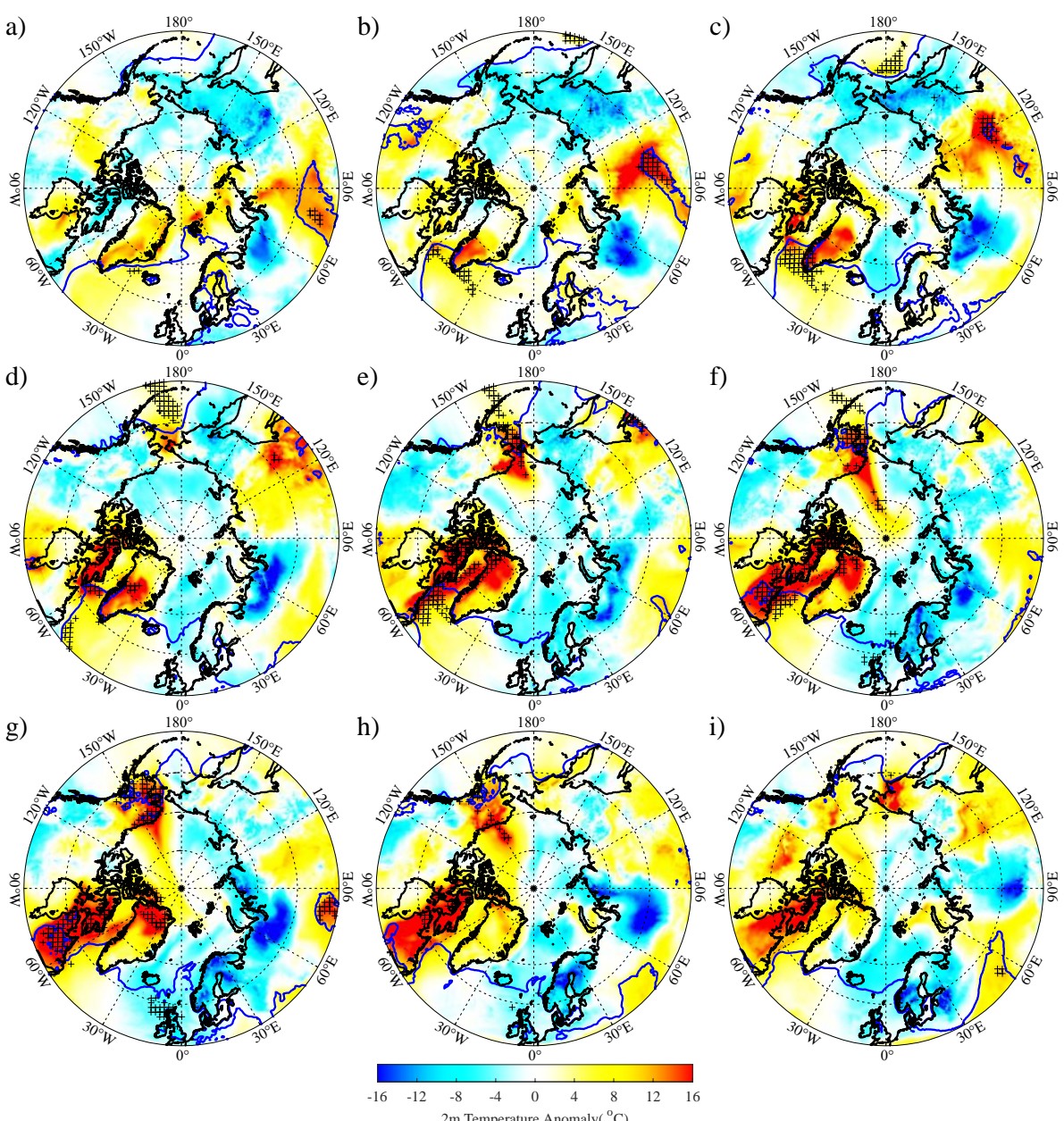

**Figure 7.** Two-meter air temperature anomaly (°C) from the ERA5 at 0 GMT on: a) 2 March, b) 3 March, c) 4 March, d) 5 March, e) 6 March, f) 7 March, g) 8 March, h) 9 March, and i) 10 March 2023. The anomalies are shown with respect to the period 16 February – 15 March 1979-2023. Grid points where the anomalies are less than the 1st percentile or greater than the 99th percentile based on the above period are indicated with the '+'. The blue curves represent the 0°C isotherm.


Despite the 31 km resolution of the ERA5 fields, the array of synoptic maps makes it challenging to ascertain the extent of the
temperature extremes, especially along coastal areas and along the approximate ice edges. The daily T2m fields are therefore
supplemented with weather station time series to provide additional perspective on the air temperatures. During the Labrador
Sea ice loss event, above-average air temperatures at Nuuk, Greenland to the southeast of the ice edge were recorded with
>0°C daily mean temperatures from 2-7 March with warm air temperature extremes observed on 3-4 March (**Figure 8a**).

Likewise, above-freezing, extreme air temperatures were observed in the GrIS lower ablation zone in the Qassimiut region
(QAS_L) and on a glacier tangential to the Nuuk DMI station (NUK_K) during this period (**Figure S2a,b**). Meanwhile, in
Aasiaat, Greenland roughly ~500 km north of Nuuk, the air temperatures were above-normal during this time, but were not
above-freezing or considered extreme by the criteria used here (**Figure 8b**). Over the Bering Sea, St. Paul Island observed a
stint of above-freezing temperatures that ranked near the 99[th] percentile for 4-7 March (**Figure 8c**), while Kotzebue on Alaska's
northwest coast saw near- to slightly-above normal air temperatures during the Bering ice loss period but the airstream neither
exceeded 0°C nor the 99[th] percentile criteria (**Figure 8d**).


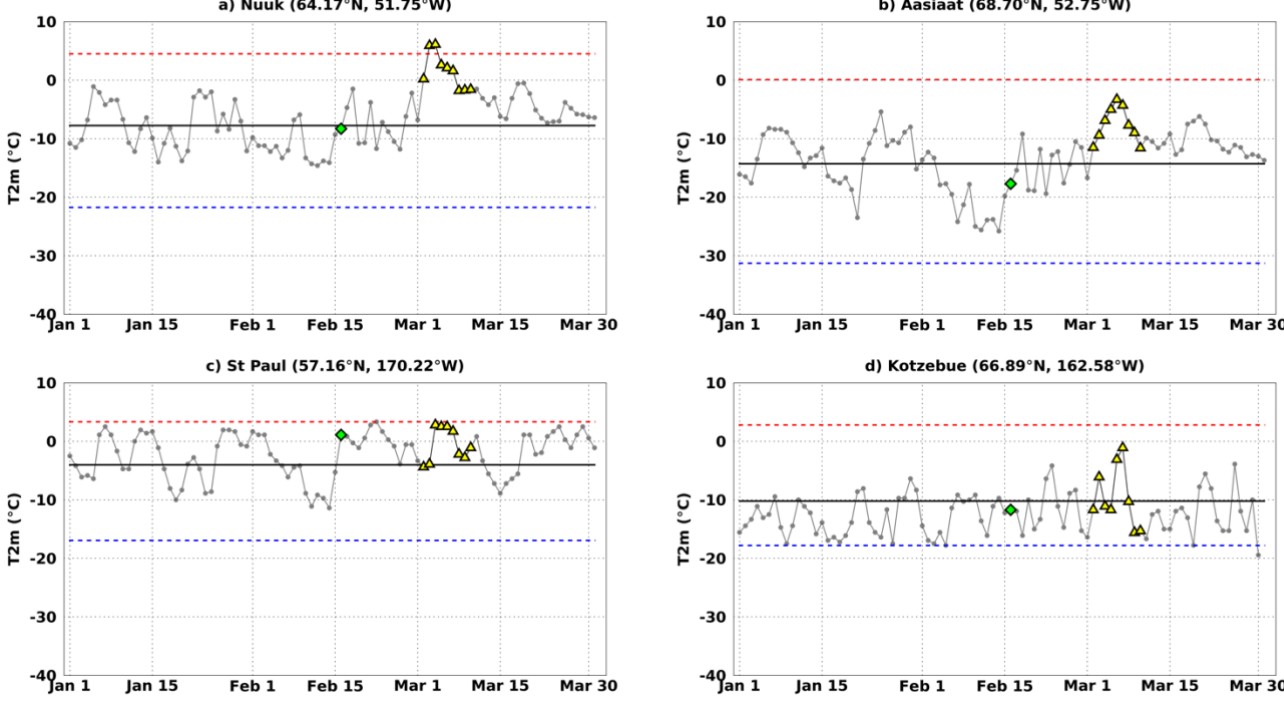

**Figure 8.** Weather station two-meter air temperature (°C) 1 January – 31 March 2023 daily time series (gray lines) overlapping the multi-sectoral melt event for a) Nuuk, b) Aasiaat, c) St. Paul, and d) Kotzebue. Considering all days from 1 January to 31 March for the respective stations' full periods of record (see Section 2.1), the mean T2m (black line), 1st percentile (blue dashed line), and 99th percentile (red dashed line) are shown in each graphic. The sudden stratospheric warming event on 16 February 2023 is labelled with a green diamond, and to draw attention to the dates around the Labrador Sea and Bering Sea melt events, the period from 2-10 March 2023 is identified by yellow triangles. For reference the weather stations are overlaid on **Figure 1**.

Further analysis into the thermodynamic environment revealed that the anomalously warm airstreams advected over both the
Labrador and Bering regions possessed extreme water vapor content around the time of their respective melt peaks shown in
time series in **Figure 9a,b** and **Figure 10a,b** and in maps presented in **Figure S3**. During these peaks, both seas experienced
anomalous net and downwelling radiation in excess of the 95th percentile (**Figures 9c,d** and **Figures 10c,d**) with that energy
likely driving ice loss through melt. To further investigate the hydrometeorological nature of these airstreams the Guan and
Waliser (2019) atmospheric river (AR) detection algorithm was run separately for the Labrador Sea and Bering Sea domains
shown in **Figure 1**. Warm, moist conditions that overlapped these melt events were associated with AR activity (**Figure 11**).
An AR resided over the Labrador Sea for >40% of the day on 3-4 March, and its residence time was extreme on 5 March
(~60% of the day; **Figure 11a**). Moisture within this AR (**Figure 11c**) and total precipitation from the AR (**Figure S4**) were
both above-average, but not extreme. Meanwhile, daily AR residence time within the Bering Sea exceeded 40% on 4-7 March,
with an AR duration extreme (>60% of the day) on 5 March preceding the short-term melt peak on 6 March when IVT was
also extreme (**Figure 11b,d**). Extreme ERA5 daily precipitation associated with the AR intrusion fell in the Bering region
from 5-8 March (**Figure S4**). These persistent and anomalously warm and wet airmasses contributed to these tandem melt
extremes. Further analysis is ongoing to examine the full surface energy budget, including the role of rainfall, toward shaping
the observed melt.

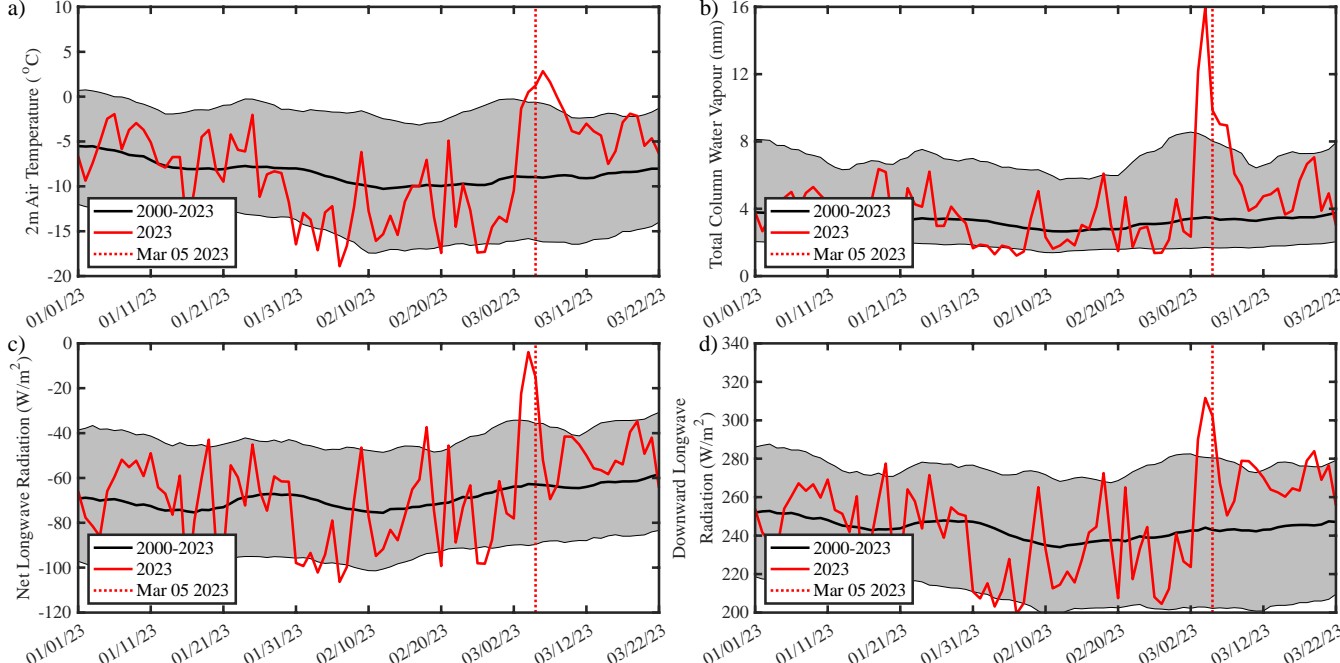


**Figure 9.** Time series (red curves) of ERA5: a) two-meter air temperature (°C), b) total column water vapor (mm), c) net
longwave radiation (W/m$^2$), and d) downward longwave radiation (W/m$^2$) averaged over the Labrador Sea region, indicated
in Figure 1b, for the period January 1 to March 26, 2023. The black line represents the climatological mean value for the period
2000-2023 with shading incorporating values between the 5th and 95th percentiles. The ending date for the 4-day window with
the largest change in sea ice concentration is shown with the dotted red line.


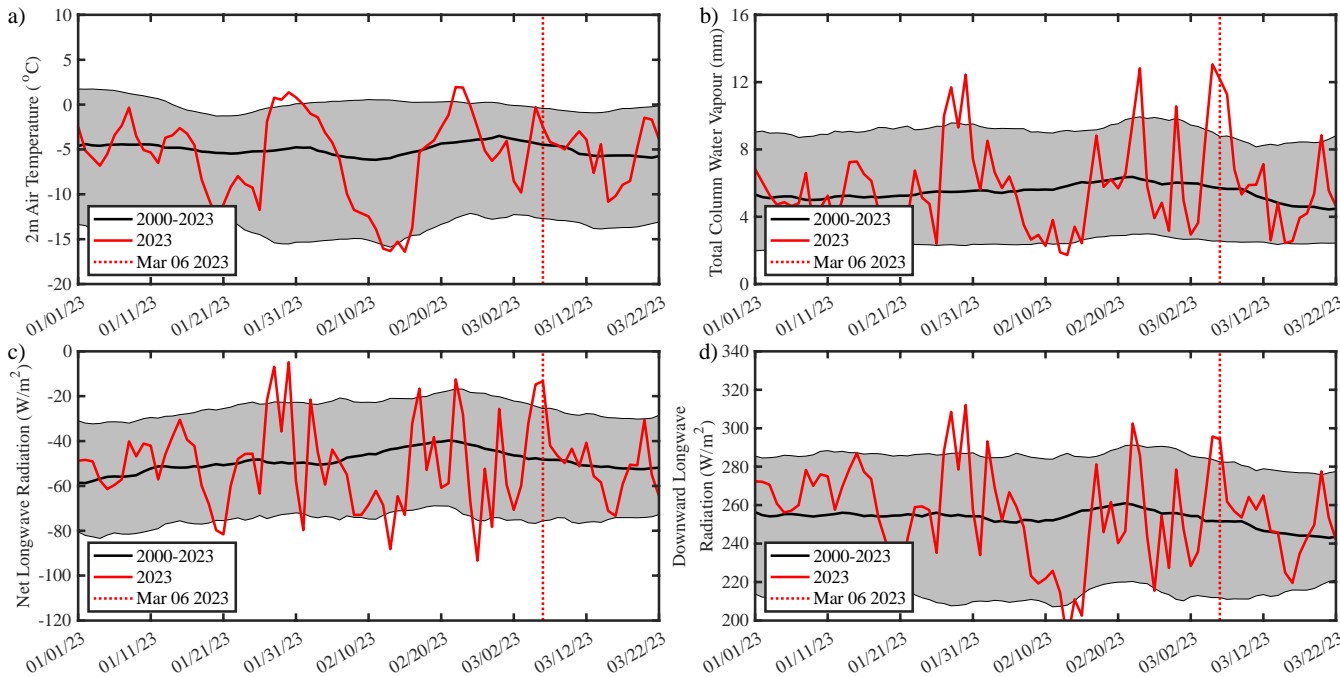

**Figure 10.** Time series (red curves) of ERA5: a) two-meter air temperature (°C), b) total column water vapor (mm), c) net longwave radiation (W/m²), and d) downward longwave radiation (W/m²) averaged over the Bering Sea region, indicated in Figure 1e, for the period January 1 to March 26, 2023. The black line represents the climatological mean value for the period 2000-2023 with shading incorporating values between the 5th and 95th percentiles. The ending date for the 4-day window with the largest change in sea ice concentration is shown with the dotted red line.

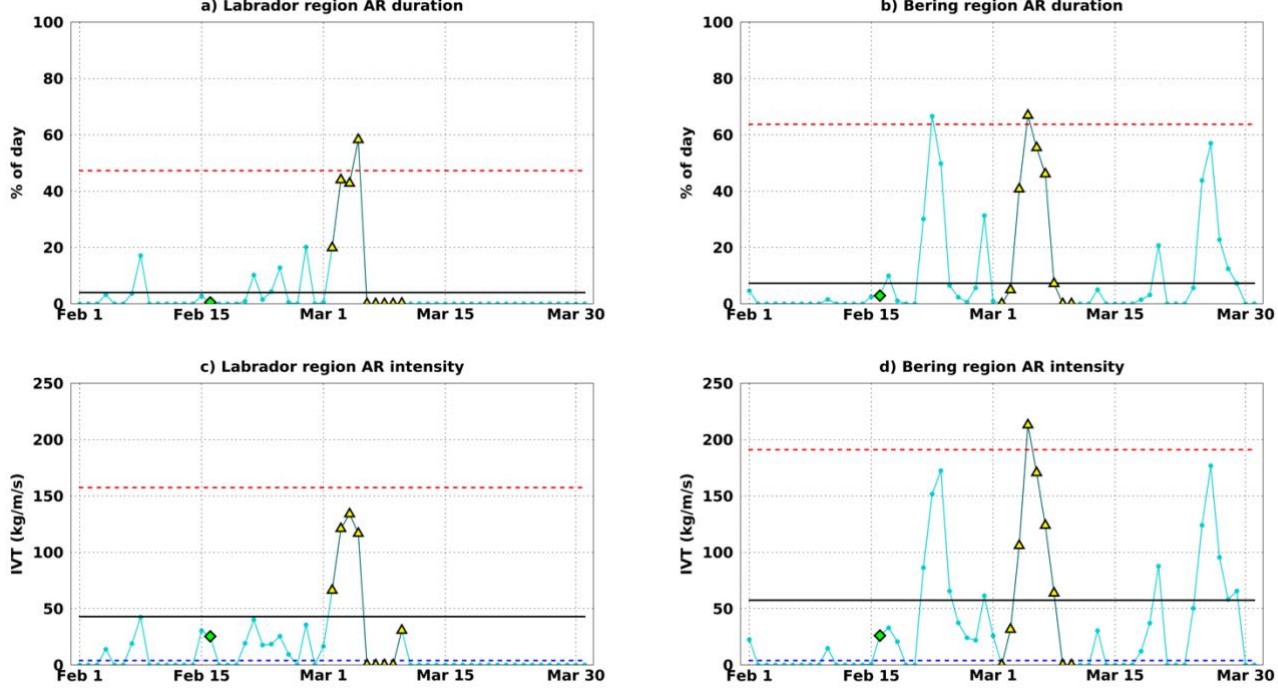

**Figure 11.** March atmospheric river (AR) duration (% of day AR in domain) and intensity (kg/m/s) for the Labrador region (a,c) and Bering region (b,d), respectively (teal lines). The AR data are calculated over the same domains as shown in **Figure 1**. The thick black line in each panel represents the 1979-2023 mean. Considering all days from 1 February to 31 March for the respective regions for the 1979 to 2023 period, the 99th percentile (red dashed lines) are shown in all panels while the 1st percentile represents AR non-occurrence, and therefore is not marked in these plots. The sudden stratospheric warming event on 16 February 2023 is labelled with a green diamond, and to draw attention to the dates around the Labrador Sea and Bering Sea melt events, the period from 2-10 March 2023 is identified by yellow triangles.

## 4. Discussion and conclusions

Tandem, unusually-timed sea ice melt extremes in the Bering Sea and Labrador Sea occurred in early March 2023. The retreat
of the ice edge in both marginal seas was similarly driven by the confluence of anomalous meteorological phenomena. Mid-
tropospheric heights increased and intense ridging patterns developed over the Labrador Sea and Bering Sea during the time
in which the respective regional ice loss events occurred. A longitude-pressure analysis (**Figure S1**) revealed that a SSW in
February 2023 was strongly linked to the mid-tropospheric height increases over the Labrador Sea region in early March, while
the height increases over the Bering Sea were isolated to the troposphere, and were likely linked to a fortuitous shift of the
large-scale La Nina-related ridging over the North Pacific into the Alaskan region. Below we discuss the ice loss events and
focus on the attendant atmospheric mechanisms that provided thermodynamic support for their occurrence.

*4.1 Perspectives on ice losses during the maximum and supporting atmospheric processes*

Amidst the decline of winter season ice coverage and thickness in the warming Arctic, the latitude of the ice edge can vary on
daily timescales due to wind and melt-driven processes. However, the probability curves shown in **Figure 3** suggest that such
short-term March 2023 sea ice losses in either the Bering or Labrador regions, taken independently, qualify as extreme events.
Both the magnitude of losses and the unusual timing of their anomalous occurrence aligned with the climatological Arctic sea
ice maximum may further qualify these melt extremes collectively as a rare synoptic ice loss event. We do not assess ice edge
changes in other marginal seas during the March historical record to establish whether other areas participated in this event.

The anatomy of the melt extremes can be described by a confluence of anomalous atmospheric phenomena that simultaneously
occurred over the Bering Sea and Labrador Sea. The melt period was preceded by an SSW event that led to a shift in the large-
scale mid-tropospheric circulation regime over the polar cap as evidenced by the rapid transition over two weeks from strong
positive to negative NAO conditions and lower to higher mid-tropospheric air pressure over the high Arctic, in particular over
Greenland. The noted shift to negative NAO followed by the development of a Greenland block that supported southerly winds
and warm advection across the Labrador Sea following a SSW has been documented in previous studies (e.g., Charlton-Perez
et al., 2018; Domeisen, 2019; Domeisen and Butler, 2020). While the set-up of the Greenland block is not unique to this event,
its magnitude for the time of year is remarkable as shown by the extremes highlighted in the GBI time series (**Figure 5c**) and
z500 spatial plots (**Figure 6**).

SSWs on average tend to elicit a weaker atmospheric dynamical response over the Bering region than the Labrador Sea. Smith
et al. (2018) analyzed data from the Whole Atmosphere Community Climate Model of NCAR's Community Earth System
Model and found that over the 40 days following SSW onset there were minimal sea-level pressure (SLP) changes over the
Bering Sea and greater Alaska, but there were large, positive SLP anomalies located northward and eastward of these areas
including around Greenland. Across SSW winters (JFM), the authors also found similar SLP signatures over Greenland, but
negative SLP anomalies and northerly winds over Alaska and the Bering Sea. However, if we consider only SSWs that occur
during La Niña winters, the large-scale circulation response following these events (**Figure S5b**) looks very similar to the
patterns seen in 2023 (**Figure S5a**), with ridging over both Greenland and the Aleutians. The interpretation is that the SSW
drives most of the tropospheric height changes over Greenland and the North Atlantic, while La Niña background conditions
favor North Pacific ridging into the Gulf of Alaska. In addition to the SSW event and La Niña phase, factors such as internal
variability of the climate system and air-sea interactions over the North Pacific Ocean may have played a role in inducing the
anomalously strong mid-tropospheric ridge extending from Greenland to Alaska.

In both the Labrador Sea and Bering Sea, anomalous atmospheric circulation characteristics, namely the stationary, extreme
blocking anticyclones, supported southerly advection of above-normal to extremely warm and moist air that led to these
thermodynamically-driven melt events (e.g., **Figures 7, 9-11,** and **Figure S2**). Additional investigation of the airstreams
revealed that anomalous ARs were present in both regions during this time and played a critical role in the simultaneous melt
extremes. The extreme duration of the AR over the peak Labrador Sea melt and extreme duration and intensity immediately
preceding the Bering Sea melt, both on 5 March, likely enhanced downwelling longwave energy transfer into the ice, causing
its short-term, yet remarkable, decline. Past studies have likewise identified downward longwave radiative flux during AR
passage as a key process that tends to decrease ice mass balance during summer (Mattingly et al., 2018; Wille et al., 2019;
Francis et al., 2020) and slows wintertime ice growth (Hegyi and Taylor, 2018; Zhang et al., 2023). Here, we document the
role of anomalous large-scale meteorological characteristics, including tandem AR events, that drove unprecedented and
concurrent sea ice melt at a time of year characterized by maximum ice extent.

*4.2 Additional considerations emanating from this case study*
This rare ice loss event concurrently encompassing the Bering Sea and Labrador Sea was shaped by a confluence of synoptic
extremes that aligned in time to induce thermodynamic melt of the sea ice edge. We look at this ice loss from a thermodynamic
perspective, though concede that in addition to supporting melt that southerly winds could have induced some sea ice
compaction in the marginal ice zones through convergence. If this event was examined through a sea ice budget lens, we
acknowledge that producing estimates of ice dynamical processes, such as wind-driven convergence and divergence, would
be important to gain a more complete understanding of the evolution of mechanisms responsible for these regional ice losses.
Follow-on work will take a broader view of thermodynamic processes, which may provide additional insight into ice loss
mechanisms elucidated in this case study. For example, resolution of the sea ice types and surface energy balance before,
during, and after the melt event may provide perspectives on ice-air interactions that shaped it.

Related to the surface energy balance processes, further analyses will delve deeper into the roles of latent heating and humidity
fluxes in shaping the ice melt event. Rainfall (<1 mm) was observed during 2-3 March in the rain gauges at the Nuuk and
Aasiaat DMI weather stations, and, if it were not for sporadic station outages from 2-10 March, rain on other days during this
period may have been documented (C. Drost Jensen 2024, personal communication). Nearby, separate near-coastal weather
stations maintained by Asiaq Greenland Survey also documented small amounts (<1 mm) of rainfall at Nuuk and Kobbefjord
(A. Ginnerup 2024, personal communication). Meanwhile, terrestrial weather stations at Kotzebue and Nome, Alaska, ~300
km to the southwest, saw >25 mm of cumulative rainfall during 4-6 March, which are 3-day total precipitation records for both
weather stations in March (R. Thoman 2023, personal communication). Spatial patterns of ERA5 total precipitation over this
period are consistent with these observations (**Figure S4**). In addition to rain measurements near the coast, rain on cold snow
was also detected in weather station observations found in the southwestern GrIS accumulation zone, which is rare for the time
of year (J. Box 2024, personal communication). Further diagnostic evaluation is needed to determine the extent, frequency,
amount, and impacts of rainfall on the cold snow cover on the GrIS and sea ice during this period. Thus, follow-on studies of
the surface energy exchange processes and precipitation characteristics may help to broaden our perspective of this complex
extreme event.

It is clear from recent years that there are occurrences of a variety of extreme Arctic events that vary in location, season, and
type which meet or exceed previous records (Walsh et al. 2020). Philosophically, it is difficult to project let alone interpret the
future frequency of these events without detailed historical analogues. It has been proposed that the recent increase of Arctic
extremes is due to an overlap of steadily increasing Arctic warming that is constructively superimposed on the natural range
of atmospheric and oceanic dynamics, e.g., jet stream meanders, atmospheric blocking, storms, and upper-ocean heat content
(Overland 2022), which could themselves, at least in some cases, be influenced by anthropogenic global heating. This is
certainly the case with the concurrent examples from the Labrador Sea and Bering Sea in March 2023. Whether this extreme
event foreshadows a more frequent occurrence of similar events in the future is an open but intriguing question that merits
careful future investigation.

*Data availability*. Alaska weather station data are available from https://xmacis.rcc-acis.org. Greenland coastal weather station
records were obtained from Caroline Drost Jensen (DMI). PROMICE observations are from
https://dataverse.geus.dk/dataset.xhtml?persistentId=doi:10.22008/FK2/IW73UU. The NAO index was downloaded from
https://www.cpc.ncep.noaa.gov/products/precip/CWlink/pna/nao.shtml. ERA5 reanalysis fields are obtained from the
Copernicus Climate Data Store at https://cds.climate.copernicus.eu/cdsapp#!/dataset/reanalysis-era5-single-
levels?tab=overview. Sea ice data are downloaded from NSIDC at https://nsidc.org/data/g02202/versions/4. The SSW
Compendium can be found at https://csl.noaa.gov/groups/csl8/sswcompendium/majorevents.html.

*Author contributions*. T.B. and G.W.K.M. conceived the study with input from Q.D., A.H.B., J.E.O., R.L.T., I.B., Z.L., and
E.H. as the study developed. R.L.T. provided assistance with data acquisition. All authors provided feedback on draft iterations
of the paper.

*Competing Interests*. None.

*Acknowledgements*. T.J.B. and Q.D. were funded by NSF Arctic System Science awards 2246600 and 2246601, respectively.
J.E.O. is supported by NOAA's GOMO Arctic Research Program. PMEL contribution #53XX. GWKM was funded by the
Natural Sciences and Engineering Research Council of Canada.  E.H. was supported by NERC NE/W005875/1. The authors
wish to thank Caroline Drost Jensen and Anders Ginnerup for assistance obtaining and interpreting the Greenland weather
station records. We also thank numerous scientists for discussions on topics related to high-latitude precipitation and GrIS
meteorology, including Jason Box, Jakob Abermann, Matthew Sturm, and Melinda Webster.

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
