# Peer review of "Concurrent Bering Sea and Labrador Sea ice melt extremes in March"

_EGUsphere, 2024_

## Author Comment (AC2)

9 August 2024

Dear Reviewer 1,

We appreciate your time reviewing and providing constructive feedback on our paper. We have considered and responded to each comment offered, and our responses follow below in **bold**. In addition to manuscript revisions stemming from your comments, we have also slightly modified the title and some of the wording of the text to reflect the nature of the "concurrent" Arctic extremes we are documenting. A new paragraph has been added toward the end of the Introduction section to better frame our study around this relatively new research space emerging on concurrent weather extremes.

Thank you again for your constructive review of our paper.

Sincerely,
Tom Ballinger
Corresponding Author
* * *
RC1 Comments

The authors investigated the effects of the SSW and La Nina teleconnections on the unseasonal melt events of the tandem ice loss over the Bring and Labrador Seas in March 2023. Associated large-scale anticyclonic anomalies funneled warm and atmospheric rivers to the bi-regional ice melt events. These results are generally interesting, focusing on the combination of stratospheric and La Nina-related tropospheric effects on Arctic surface thermal conditions. However, the atmospheric circulation features and characteristics associated with La Nina are rather unclear. I would like to propose further research into the effects of La Nina on the Arctic.

**Thank you for your thoughtful remarks on our paper. We agree the mentioning of La Niña in the Abstract and text in the absence of more detailed analysis was a rather hasty mention at best. We elaborate further on this teleconnection's links with Arctic sea ice melt in our response to your second point below.**

1.  It is quite clear that the SSW occurred around 15 Feb and then propagated downward to the surface around 6 Mar, and the Greenland and Alaska blocking intensified. The first question is, how can we link the SSW to the increased mid-tropospheric blocking? Could the authors provide more evidence of the evolution and pattern of the polar vortex?

**To better illustrate the connection between the SSW and the mid-tropospheric blocking, Figure R1 shows the 45-75°N eddy geopotential height anomalies (Z\*) as a function of pressure vs longitude (where Z\* is calculated as the deviation from the zonal-mean), for the periods March 2-6 and March 7-11 2023 (using ERA5 reanalysis data; anomalies are calculated relative to the 1979-2023 climatology). The influence of the SSW shows up as anomalously positive Z\* from the mid-stratosphere to the surface over the**

**Greenland/North Atlantic sector (this feature begins to extend downward from the stratosphere towards the end of February; not shown here, but matches the timing in Figure 4). Also evident is the persistent tropospheric ridge over the N. Pacific/Alaskan region, which amplifies and extends vertically during the March 7-11 period.**

[Figure]

**Figure R1.** The 45-75°N eddy geopotential height anomalies as a function of pressure vs longitude. These anomalies are calculated as the deviation from the zonal-mean for the periods relative to the 1979-2023 climatology for March 2-6 (top) and March 7-11 2023 (bottom).

2. The La Nina teleconnections are highlighted in the abstract and the main paper, but the associated atmospheric circulation and physical mechanisms are missing. I do not think the present evidence is sufficient to guarantee such a causal link between La Nina and Arctic sea ice melt.

**We agree with the reviewer's comment and have removed mention of La Niña from the Abstract as our primary analyses shown in the paper do not explicitly analyze or compare the winter sea ice response of 2023 to past ice cover states during La Niña events. That said, we elect to keep the brief mention of La Niña as a potential background forcing mechanism of ridging atop Alaska in Section 3.2 as a prelude to elaborating more within our Discussion in Section 4.2 on La Nina's role in shaping this upper-level feature. In the latter section, Figure S3 is referenced which compares z500 and T2m conditions in the 2023 SSW under La Niña conditions (Figure S3a) to those same fields during other such La Niña phases of similar winter periods in 1984, 1989, 1999, 2001, 2006, 2008, 2009, and 2018 (Figure S3b). While explicit links to between tropical Pacific-origin teleconnections and Arctic sea ice are not focused upon in our study, we feel it is important to include mention of links between past La Niña events and upper-level circulation and surface temperatures in the Arctic as compared with the 2023 case we focus upon.**

3. Surface air temperature anomalies reach up to 15K over the Labrador and Bering Seas, but the extent and area of sea ice melt is quite small. Could the authors quantify the relative changes in area, extent and concentration?

**It is true that despite the warm surface air temperature anomalies of ~15K that persisted over the Labrador and Bering Seas concurrently from 5-7 March (Figure 7d-f), the sea ice concentration (SIC) fluctuations in the respective marginal seas are not all that extreme (Figure 2). We believe showing the area and extent series would not change this argument. What stands out, however, are the large 4-day changes in early March SIC that occurred concurrently in both areas (Figure 3). These concurrent SIC loss features and their complex set of atmospheric drivers is what we focus upon in the paper.**

**On a related note, the idea of "concurrent extremes" is one gaining traction in the literature. As such we elect to replace "simultaneous" with "concurrent" in the title and several places in the text. We have added a paragraph on the nature of such events in the second to last paragraph of the Introduction section in an effort to place this study in line with papers recently published on this topic.**

4. The atmospheric river plays an important role in Arctic warming in March. I am curious about the atmospheric water vapour transports associated with the blockings and the role of water vapour on surface temperature. Is the water vapour converted into snow or rain to lower the temperature, or is the surface warmed by increased longwave radiation?

**Observations suggest that rain and increased longwave during atmospheric river passage may have impacted the melt events. Rain was measured in multiple rain gauges in coastal observations near the ice pack edge (Nuuk, Kobbefjord, and Aasiaat in west Greenland and Kotzebue and Nome in northwest Alaska). As mentioned in the second to last**

**paragraph of Section 4.2, a surface energy budget (SEB) analysis including the contribution of latent heating and humidity fluxes into the melt anomalies is subject of follow-on study.**

**In lieu of SEB analyses over both regions, we have made new winter-long new time series plots of two-meter air temperature, total column water vapor, net longwave (LW) radiation, and downwelling LW over the Labrador Sea and Bering Sea. These are shown below and have been inserted, along with accompanying text, into the revised paper as new Figures 9 and 10, respectively. These plots clearly show how the respective peak four-day melt events coincide within +/-1 day of maximum temperature, moisture, and net/downwelling LW radiation in the respective areas. We comment on these new findings in the last paragraph of Section 3.3.**

[Figure]

**Figure 9.** Time series (red curves) of ERA5: a) two-meter air temperature (°C), b) total column water vapor (mm), c) net longwave radiation (W/m²), and d) downward longwave radiation (W/m²) averaged over the Labrador Sea region, indicated in Figure 1b, for the period January 1 to March 26, 2023. The black line represents the climatological mean value for the period 2000-2023 with shading incorporating values between the 5th and 95th percentiles. The ending date for the 4-day window with the largest change in sea ice concentration is shown with the dotted red line.

[Figure]

**Figure 10.** Time series (red curves) of ERA5: a) two-meter air temperature (°C), b) total column water vapor (mm), c) net longwave radiation (W/m$^2$), and d) downward longwave radiation (W/m$^2$) averaged over the Bering Sea region, indicated in Figure 1e, for the period January 1 to March 26, 2023. The black line represents the climatological mean value for the period 2000-2023 with shading incorporating values between the 5$^{th}$ and 95$^{th}$ percentiles. The ending date for the 4-day window with the largest change in sea ice concentration is shown with the dotted red line.

5. Figure 5: I would suggest that the authors show the climatological daily evolution to facilitate contrasts between the extreme event and the climatology, so that the magnitude of the anomalies is more apparent. I am also curious about the magnitude of the geopotential height and 2m temperature, which reach 500m and 16k respectively. Is this correct?

**Below we have included the daily means (orange lines in Figure R2 below) for each variable's full period of record for reference. As there is minimal difference between these values and the full winter period averages from 1 January to 31 March (solid black lines) as shown in existing Figure 5, we prefer to keep the time series as presented, which are consistent with Figures 8 and 9.**

**In terms of the second comment, we checked the data and the z500 and two-meter air temperature anomalies are correct as plotted in Figures 6 and 7, respectively.**

[Figure]

**Figure R2.** Daily atmospheric indices for 1 January – 31 March 2023 (purple lines) overlapping the multi-sectoral melt event for the a) Polar Vortex Index (m/s), b) North Atlantic Oscillation Index (standardized), c) Greenland Blocking Index (m), and d) Alaska Blocking Index (m). Considering all days from 1 January to 31 March for the respective indices full periods of record (see Section 2.1), the mean of each variable (black line), 1st percentile (blue dashed line), and 99th percentile (red dashed line) are shown in each graphic. Daily means are also shown by the solid orange lines. The sudden stratospheric warming event on 16 February 2023 is labeled with a green diamond, and to draw attention to the dates around the Labrador Sea and Bering Sea melt events, the period from 2-10 March 2023 is identified by yellow triangles.

---

## Author Comment (AC3)

9 August 2024

Dear Reviewer 2,

Thank you for your time reviewing and providing constructive feedback on our paper. We have considered and responded to each comment offered, and our responses follow below in **bold**. In addition to manuscript revisions motivated by your comments, we have also slightly modified the title and some of the wording of the text to reflect the nature of the "concurrent" Arctic extremes we are documenting. A new paragraph has been added toward the end of the Introduction section to better frame our study around this relatively new research space emerging on concurrent weather extremes.

Thank you again for your review of our paper.

Sincerely,
Tom Ballinger
Corresponding Author
* * *
RC2 Comments

This study examines the rare melting events in the Labrador Sea and Bering Sea in March 2023, a time when the sea ice area typically reaches its maximum during the boreal winter. These melting events are explained by intense moisture transport from lower latitudes, driven by anomalous blocking over high latitudes. The authors also attribute the circulation anomalies to a preceding sudden stratospheric warming (SSW) event and its downward influence. As a case study, the authors effectively link the surface ice melting to stratospheric extremes. However, I have some suggestions and comments regarding the manuscript, as detailed below:

1. I agree with another reviewer's comments on the role of La Niña. The authors did not provide evidence to support this argument. They should either add more evidence regarding La Niña or remove the related conclusion. Considering the manuscript's length, it would be better to remove the arguments about La Niña from the abstract and conclusion. It is still acceptable to mention La Niña among other factors that may have played a role.

**We agree with both reviewers' comments. As mentioned in our response to comment #2 of RC1, we elect to keep the brief mention of La Niña as a potential background forcing mechanism of ridge development atop Alaska in Section 3.2, which then dovetails to a more elaborate Discussion in Section 4.2 on La Nina's role in shaping blocking anticyclone centers over the regions that experienced concurrent sea ice melt.**

2. All figures need improvement. The words/characters are too small.

**We have enlarged all the figures and recreated them in high-resolution formats to improve their legibility and interpretability.**

3. To what extent these two events are extreme? It would be helpful to show a line to denote the 5(1)% percentile or -1.5(-2) sigma.

**The rarity of these short-term, multi-day melt events is highlighted in Figure 3. This histogram indicates that the 20% four-day decline in SIC over both the Labrador Sea (red dashed line) and Bering Sea (blue dashed line) is rare (<1st percentile events for March since 2000).**

4. Figure 5 caption: The authors mentioned the black line denoting the mean T2m. Is this an error? It seems to represent the mean values of these variables.

**Thanks for catching our error. We have altered this to say this is the mean for each variable.**

5. It would be great to reorganize the size of the subplots in Figures 6 and 7. The current subplots are too small and not reader-friendly.

**We have slightly modified the size of figures to make both the fields and corresponding text more legible. Final uploaded figures in eps and tif formats are more legible than the version of the paper under review.**

6. Lines 35-36: Add Zhang et al. 2018 (10.1126/sciadv.aat6025).

**Thanks for your recommendation. We have added the suggested reference.**

7. Lines 148-149: Related to comments #3. The authors argued that the melting event is 'unprecedented.' However, the SIC anomalies 'did not breach the 5th or 95th percentiles' and thus do not seem that extreme. Could the authors re-examine the calculation or further clarify?

**As we mention in response to the reviewer's third comment (as well as the third comment of RC1), while the sea ice concentration (SIC) fluctuations in the respective marginal seas are not all that extreme as shown in Figure 2, the 4-day change in early March SIC that occurred concurrently in both areas is extreme (Figure 3). This feature and the meteorological conditions that shaped it comprise the foci of our study.**

8. Lines 314-315: The authors did not show any evidence about the downward longwave radiation or surface energy balance in the current study. Given this is in the discussion part, it would be better to cite previous studies here. Some polar AR studies have already presented these results, such as Hegyi and Taylor 2018 (10.1029/2017GL076717), Mattingly et al. 2018 (10.1029/2018JD028714), Wille et al. 2019 (10.1038/s41561-019-0460-1), Frances et al. 2020 (10.1126/sciadv.abc2695), Zhang et al. 2023 (10.1038/s41558-023-01599-3), etc.

**We have added some analysis on downward longwave radiation during the AR events and added the suggested references within the revised last paragraph of Section 4.1.**

---

## Author Response (AR2)

25 September 2024

Dear Reviewer 3,

We appreciate your time reviewing and providing constructive feedback on our revised paper. We have considered your remarks, and our responses are found below in **bold**.

Thank you again for your constructive review of our paper.

Sincerely,
Tom Ballinger
Corresponding Author
* * *
RC3 Comments

Review of "Concurrent Bering Sea and Labrador Sea ice melt extremes in March 2023: A confluence of meteorological events aligned with stratosphere-troposphere interactions"

I did not review this paper in its first round, however the editor asked me to offer my thoughts on the revised submission. I will focus my comments on the stratosphere-troposphere coupling parts of the paper. (I do not consider myself qualified to review the other aspects of the paper, and will not include them in my review.)

The language used to associate the melt events with SSWs is too strong. There is strong evidence that blocking near Greenland is indeed a byproduct of SSWs, however there is no evidence that SSWs are typically followed by blocks (or any robust climate anomalies for that matter) near the Bering Sea. There is actually a reasonably large literature on whether (if at all) SSWs are followed by a Pacific sector impact (e.g. Dai et al 2024 and references therein). There is no robust response in reanalysis data. However many models simulate a response (though clearly not a block as was observed in March 2023) for reasons that are still not fully understood.

To some degree the authors note this in the paragraph starting near line 320, however there are some additional papers that they might consider citing in this paragraph listed below.

Given this, the strongest claim that can be made is that the block over Greenland is likely associated with the SSW, however the block over Alaska had nothing to do with it, and may have instead been associated with the La Nina event. To be specific, I think the language used near line 63, 237-238, and 299 is too strong, even as the paragraph starting near line 320 is reasonable.

Dai, Y., P. Hitchcock, and I. R. Simpson, 2024: What Drives the Spread and Bias in the Surface Impact of Sudden Stratospheric Warmings in CMIP6 Models?. J. Climate, 37, 3917–3942, https://doi.org/10.1175/JCLI-D-23-0622.1.

Zhang, J., Tian, W., Pyle, J.A., Keeble, J., Abraham, N.L., Chipperfield, M.P., Xie, F., Yang, Q.,

Mu, L., Ren, H.L. and Wang, L., 2022. Responses of Arctic sea ice to stratospheric ozone depletion. Science Bulletin, 67(11), pp.1182-1190.

Liang, Yu-Chiao, Young-Oh Kwon, Claude Frankignoul, Guillaume Gastineau, Karen L. Smith, Lorenzo M. Polvani, Lantao Sun et al. "The Weakening of the Stratospheric Polar Vortex and the Subsequent Surface Impacts as Consequences to Arctic Sea Ice Loss." Journal of Climate 37, no. 1 (2024): 309-333.

Kelleher, Michael E., Blanca Ayarzagüena, and James A. Screen. "Interseasonal connections between the timing of the stratospheric final warming and Arctic sea ice." Journal of Climate 33, no. 8 (2020): 3079-3092.

**We appreciate the reviewer's perspectives as fair points are made about toning down the language in various places involving SSW impacts on subsequent high-latitude tropospheric circulation features. We have made an effort to do this through the revised manuscript as recommended. Moreover, in revised Section 3.2 and Section 4 we explicitly note that following SSWs that blocking anticyclone formation over the Greenland/Labrador region, such as occurred in March 2023, is typical (e.g., Baldwin et al., 2021). However, the development of mid-tropospheric ridging patterns in and around the Alaska region during this time is more likely to be a signature of the La Niña background state than in response to the SSW event.**

**Rather than include the reviewer's literature recommendations into the revised text, we bring attention to Section 4 modifications around new Figure S1 (see supplemental material and below). As stated in the first paragraph of this section, "A longitude-pressure analysis (Figure S1) revealed that a SSW in February 2023 was strongly linked to the mid-tropospheric height increases over the Labrador Sea region in early March, while the height increases over the Bering Sea were isolated to the troposphere, and were likely linked to a fortuitous shift of the large-scale La Nina-related ridging over the North Pacific into the Alaskan region." We believe these changes more accurately reflect the confluence of processes that shaped this concurrent event.**

[Figure]

**Figure S1.** The 45-75°N eddy geopotential height anomalies (m) as a function of pressure versus longitude. These anomalies are calculated as the deviation from the zonal-mean for the periods relative to the 1979-2023 climatology for a) March 2-6 and b) March 7-11 2023.